# A Population-Based Long-Term Follow-Up of Soft Tissue Angiosarcomas: Characteristics, Treatment Outcomes, and Prognostic Factors

**DOI:** 10.3390/cancers16101834

**Published:** 2024-05-11

**Authors:** Christina Enciso Holm, Mathias Ørholt, Maj-Lis Talman, Kiya Abebe, Andrea Thorn, Thomas Baad-Hansen, Michael Mørk Petersen

**Affiliations:** 1Department of Orthopedic Surgery, Rigshospitalet, University of Copenhagen, Blegdamsvej 3, 2100 Copenhagen, Denmark; andrea.pohly.jeppesen.thorn@regionh.dk (A.T.); michael.moerk.petersen@regionh.dk (M.M.P.); 2Department of Plastic Surgery and Burns Treatment, Rigshospitalet, University of Copenhagen, Blegdamsvej 3, 2100 Copenhagen, Denmark; mathias.oerholt.nielsen@regionh.dk (M.Ø.); kiya.abebe.01@regionh.dk (K.A.); 3Department of Pathology, Rigshospitalet, University of Copenhagen, Blegdamsvej 3, 2100 Copenhagen, Denmark; maj-lis.moeller.talman@regionh.dk; 4Department of Orthopaedic Surgery, Tumor Section, Aarhus University Hospital, Palle Juul-Jensen Blvd, 8200 Aarhus, Denmark; thombaad@rm.dk

**Keywords:** angiosarcoma, soft tissue tumors, surgery, margin, recurrence, prognostic factors

## Abstract

**Simple Summary:**

Angiosarcoma is a rare type of soft tissue sarcoma with a progressive and unpredictable disease course. Currently, surgery is the only well-defined treatment option. Adjuvant oncological treatment remains inconclusive. In effort to provide evidence for improved management and follow-up, the purpose of this study was to assess treatment outcomes and identify prognostic factors. We identified a nation-wide cohort of 154 patients with complete follow-up. Adjuvant oncological treatment did not benefit overall survival, risk of metastasis, or recurrence. Cutaneous tumors, surgery, and negative margin showed improved overall survival, yet without reducing the risk of recurrence and metastasis, resulting in a grave overall survival rate.

**Abstract:**

Angiosarcoma is a rare aggressive and understudied soft tissue sarcoma with pending evidence-based treatment guidelines due to varying study cohorts and inconsistent outcome measures. Surgery with wide resection is currently considered to be the cornerstone in management. In a population-based cohort identified from Danish National Health Registers between 2000 and 2017, this study aimed to define prognostic factors in patients with newly diagnosed soft tissue angiosarcoma. Kaplan–Meier survival analysis demonstrated 5-year overall survival of 28%. Competing risk analysis demonstrated cumulative incidence of local recurrence of 30% and metastasis of 43%. Multivariable Cox models among 154 included patients demonstrated age above 60 years and metastasis to be independently associated with worse overall survival. Cutaneous tumors, surgery, and negative resection margin were independently associated with improved overall survival. Adjuvant oncological treatment did not improve overall survival, risk of metastasis, or recurrence. Negative margin was not associated with lower risk of recurrence and metastasis. We conclude that, despite demonstrated improved survival after surgery with wide resection, overall survival remains poor.

## 1. Introduction

Angiosarcoma (AS) is a rare soft tissue sarcoma, comprising 2–5% of all soft tissue sarcomas. An increased incidence during the last forty years has been suggested, though only recently demonstrated [1]. Tumors originate from vascular cells and rarely lymphatic endothelial cells, hence arising at any anatomic site. Primary tumors arise de novo and are of mesenchymal origin, whereas secondary AS is associated with well-described external exposures. Known risk factors are radiation therapy, chronic lymphoedema (Stewart–Treves syndrome) and some familial syndromes including Maffuccis syndrome, neurofibromatosis, and Klippel–Trenaunay. By current consensus, angiosarcoma is considered an aggressive high-grade tumor with considerable risk of local recurrence and metastasis and poor overall survival rate [2,3]. Angiosarcomas are classified into subtypes depending on location and site of origin. Cutaneous tumors located at the head and neck in elderly men are often referred as the most common type of AS, whereas visceral tumors are least common [2,4,5,6,7].

Owing to its rare nature and low awareness, AS is understudied with limited evidence available in literature to guide management. The majority of studies are based on small institutional series with inconsistent reporting, leading to varying results [6,8,9,10,11]. To date, surgical management with wide resection achieving negative margins is, overall, considered as first-line choice, particularly in localized disease [5,7,12]. Adjuvant radiotherapy has been suggested to reduce local recurrence and, in some historical studies, benefit overall survival [9]. The reported results after adjuvant radiotherapy are nevertheless conflicting [2,4,6,7,9,10,11,13]. The role of chemotherapy, as well as the choice of agents, remains inconclusive [2,4,7,12].

Due to the high malignancy and limited treatment options, the overall five-year survival in patients with AS is considered to be poor, ranging from 22% to 60% [6,14,15]. The considerable variability in overall survival has been suggested to depend on factors such as primary or secondary origin and subtype, although this remains inconclusive. Prognostic factors influencing overall survival have been suggested, including radical surgery with negative margins, adjuvant radiotherapy, age, tumor size, and anatomic site. However, due to inconsistent reporting, inclusion, and follow-up time, convincing evidence-based prognostic factors are lacking [2,11,15]. Hence, current consensus management is still argued, and evidence-based clinical guidelines and follow-up recommendations are pending.

In effort to provide evidence-based recommendations for treatment and follow-up in patients with angiosarcoma, the aims of present study are (1) to describe a consecutive national cohort of patients diagnosed with soft tissue angiosarcoma; (2) to estimate long-term overall survival, risk of local recurrence, and metastasis; and (3) to analyze risk factors for overall survival, local recurrence, and metastasis.

## 2. Materials and Methods

### 2.1. The Danish National Health Registries

In Denmark, nationwide data of patients with sarcomas are available due to The Danish Sarcoma Registry [16]. The Danish Sarcoma Registry (DSR) is a prospectively maintained national database since 1 January 2009. Patients from 2000 to 2008 were later included in the DSR via validation through the Danish Cancer Registry and the Danish National Pathology Registry [17]. The registry contains information about patient characteristics, tumor characteristics, diagnostics, details on treatment, local recurrence, metastasis, as well as comorbidity and deaths. Furthermore, The Danish National Pathology Register (DNPR) comprises reports and diagnoses of all specimens assessed by Danish pathologists via data capture. Nationwide, all departments of pathology are legally obligated to report pathology data from public and private hospitals and clinics, resulting in a coverage of almost 100% [17].

### 2.2. Patients

From both registers we included all patients in Denmark with the systematized nomenclature of medicine-codes (SNOMED) for AS (M9120x, M9170x, and M9130x) and topography codes, between 1 January 2000 and 31 December 2017. According to the Danish national guidelines, all specimens suspected of being sarcoma must be consulted by a highly specialized pathologist at one of the national referral sarcoma centers. Only patients with confirmed histological diagnosis by a referral Center were included (Figure 1). Oncological data were obtained from patient files. Due to the Danish social health care system all patients diagnosed with sarcoma are entitled to free government paid treatment in a public medical care system. Hence, all patients diagnosed with a sarcoma are treated at specialized national referral centers by sarcoma surgeons and every specimen is histologically examined by a board-certified sarcoma pathologist.

### 2.3. Pathology

Morphological angiosarcomas are poorly circumscribed, infiltrative tumors with a wide range of heterogeneous morphologic and architectural patterns, from large, irregular vascular structures lined with atypical endothelial cells to solid growth with only focal vasoformation, marked nuclear pleomorphism, and a high mitotic rate. All patients were diagnosed according to the guidelines from the World Health Organization [18]. Due to wide heterogeneity, there are several differential diagnoses and a large panel of immunohistochemical staining is mostly required. Positive vascular endothelial markers, primarily CD31, CD34, and ERG, at times supplemented with vimetin, were considered diagnostic. Factor VIII, FLI-1, and vascular endothelial growth factor VEGF were used in case-by-case manner. c-MYC oncogene amplification was considered to be diagnostic in irradiation- and lymphoedema-associated tumors. According to the latest WHO guidelines, the tumor grade is no longer considered applicable to angiosarcomas [3,18]. Hence, all cases were considered to be high grade. Due to poorly demarcated and infiltrative growth without a capsule or clear border, tumor size can be difficult to determine and was not always obtained. Epithelioid hemangioendotheliomas were excluded due to having different morphology and biology. After reviewing all pathology reports, we chose to define cutaneous tumors as tumors confined to the dermis. Tumors invading subcutaneous tissue were, despite current definitions, considered as deep-seated soft tissue tumors. Inconclusive or uncertain diagnoses were reviewed by a senior pathologist from our center (MT), resulting in exclusion of further two cases. Angiosarcoma of bone and viscera were excluded (Figure 1).

### 2.4. Outcomes

The primary outcome was 5-year overall survival, defined as the probability of survival from the day of pathologyverified diagnosis to death from all causes, or the end of the study. The Danish Civil Registry [19] ensures no loss of patient survival follow-up. Two foreign patients treated in Denmark were lost to follow-up. The secondary outcome was the cumulative incidence of local recurrence and metastasis. Time to local recurrence and metastases was calculated as the time from diagnosis to first pathologically verified local recurrence or metastasis. Local recurrence and metastases were defined as pathology-reported verification of regrowth at the primary or distant site, respectively. Intralesional or excisional biopsies were not considered to be surgical procedures.

### 2.5. Statistics

Patient demographics were analyzed descriptively and tested for significance using a Chi2-test (categorical variables) and Student *t*-test (continuous variables). Confidence intervals (CI) are reported as 95%CI and *p*-values < 0.05 were considered statistically significant. The probability of overall survival was estimated using Kaplan–Meier analysis. Log-rank testing was used to compare overall survival between patient sub-groups. The Aalen–Johansson estimator was used to assess the cumulated incidence of local recurrence and metastasis, calculated using a competing risk model with death as a competing risk. Gray’s test was used to assess differences between sub-groups. To identify prognostic factors for survival, univariable analyses were performed using Cox Proportional Hazards modeling. Fine and Gray competing risk regression was performed to identify prognostic factors for recurrence and metastasis. Based on clinical relevance and significance in univariable tests (*p* < 0.05), factors were selected for multivariable regression analysis. The significance of prognostic factors is reported as a hazard ratio (HR) and 95%CI. Statistical analysis and plots were performed using R 4.2.2 (R Foundation, Vienna, Austria) software.

## 3. Results

### 3.1. Patient Demographics

In total, we included 154 patients (F/M = 98/56). The mean age was 66 years (range: 22–95). The mean follow-up was 3.6 years (range: 0 days–23 years). The mean incidence rate of AS was 1.7/1.000.000/y. Our results demonstrated a tendency towards increased incidence, especially among secondary AS (Appendix A). Most ASs were located at extremities (n = 47; 31%), followed by breast (n = 44; 29%). Primary ASs constituted the majority (n = 94; 61%) vs. secondary (n = 59; 38%). Most tumors were deep-seated soft tissue tumors (n = 114; 74%) followed by cutaneous tumors (n = 40; 26%). Secondary ASs were most often radiation induced (n = 52; 52/59; 88%) and mainly located at the breast or trunk due to prior adjuvant RT after breast cancer (n = 49; 49/59; 83%). The mean time from breast cancer diagnosis until AS was 8 years (range: 2–28 years). All but one patient with prior breast cancer (n = 48) had previous breast-conserving surgery. Eleven (11/59; 19%) secondary ASs were caused by lymphedema. Seven patients (7/59; 12%) had primary chronic lymphedema and four patients (4/59; 7%) had lymphedema due to prior RT. Fifty-four patients (54/154; 35%) received adjuvant (n = 9) or palliative chemotherapy (n = 45). Fifty-eight (58/154; 38%) patients received adjuvant (n = 28) or palliative (n = 29) radiotherapy. One patient (n = 1) received unknown radiotherapy. Nineteen (n = 19) received both palliative chemo- and radiotherapy. The baseline characteristics are demonstrated in Table 1.

### 3.2. Overall Survival

Twenty-seven patients (18%) were alive at the end of the study. The probability of overall survival was 66% (95%CI: 59–74%), 28% (95%CI: 21–35%), 19% (95%CI: 12–25%) after 1, 5, and 10 years, respectively (Figure 2). The probability of overall survival in patients with a minimum follow-up of 12 months (n = 103) was 42% (95%CI: 32–51%) and 28% (95%CI: 19–37%) after 5 and 10 years, respectively (Appendix B). We found no difference in overall survival when comparing primary and secondary AS (*p* = 0.4). Cutaneous tumors demonstrated significantly better overall survival when compared with deep-seated tumors (*p* = 0.001) (Figure 3). We found a significant difference in overall survival when comparing patients who underwent surgery vs. no surgery (*p* < 0.001). Overall survival in patients with localized disease was significantly better when compared with patients with metastases at the time of diagnosis (*p* < 0.001). Patients with later metastases demonstrated significantly worse overall survival when compared to patients with localized disease (*p* = 0.002).

### 3.3. Surgical Outcomes

The majority of all patients received surgery (n = 117; 76%). One hundred and six (106/117; 91%) patients had localized disease and eleven patients had metastasis at the time of diagnosis (11/117; 9%). Patients with AS of the breast were most likely to receive surgical treatment (42/44; 95%), followed by extremities (37/47; 79%). Wide excision with negative margin was achieved in n = 98 patients (98/117; 84%). Seventeen patients had marginal excision (17/117; 15%). Two cases could not be assessed. Of those patients with marginal excision, the vast majority of tumors were located at the head and neck (7/17; 41%). Eleven patients with marginal margins developed metastasis (11/17; 65%) and three patients developed local recurrence (3/17; 18%). Six patients (6/17; 35%) with marginal excision had metastasis at the time of surgery. The vast majority of tumors resected with marginal margin were deep seated (15/17; 88%).

### 3.4. Risk of Local Recurrence and Metastasis

Forty-eight patients had a minimum of one episode of histologically verified local recurrence. Twenty-three patients (15%) had a minimum of two episodes of local recurrence. The mean time from diagnosis to local recurrence was 1.5 years (range: 59 days–5 years). The cumulative incidence of local recurrence after 1 and 5 years was 16% (95%CI: 10–22) and 30% (95%CI: 23–38), respectively (Figure 4). Sites with the highest risk of local recurrence within 5 years, were breast (45%; 95%CI: 30–60) followed by extremities (32%; 95%CI: 19–45) and head and neck (25%; 95%CI: 10–40). Locations with the highest risk of local recurrence within 5 years were deep-seated tumors (32%; 95% CI: 23–40) followed by cutaneous tumors (27%; 95%CI: 13–40). Risk of local recurrence in patients who underwent surgery was 39% (95%CI: 30–48) and 41% (95%CI: 32–50) after 5 and 10 years, respectively. Patients who had surgery with wide resection (n = 98; 98/117; 84%) had a lower risk of local recurrence (*p* = 0.004). Twenty patients had histologically verified metastatic disease at the time of diagnosis (13%) and another forty-eight (31%) within the follow-up time. Sixty-eight patients (44%) had a minimum of one metastasis. The cumulative incidence of metastases after 1 and 5 years was 29% (95%CI: 22–36) and 43% (95%CI: 35–51), respectively (Figure 5). Sites with the highest risk of metastases were breast (55%; 95%CI: 40–70) followed by extremities (50%; 95%CI: 36–64) and trunk (35%; 95%CI: 17–52). Anatomic locations with highest risk of metastasis after 5 years were deep-seated tumors (55%; 95%CI: 45–64). Most metastases were located in the lungs, followed by lymph nodes (Appendix C). The mean time from diagnosis to metastasis was 1.1 years (0 days–8 years). We found no significant difference in the risk of metastasis, when comparing wide and marginal resection in patients with localized disease (*p* = 0.8).

### 3.5. Prognostic Factors

#### 3.5.1. Survival

In univariable analysis, factors *at the time of diagnosis* associated with poor overall survival were age > 60 years (HR = 1.94 95%CI: 1.29–2.89) and metastasis at diagnosis (HR = 3.13; 95%CI: 1.87–5.24). Cutaneous tumors were associated with improved overall survival (HR = 0.18; 95%CI: 0.05–0.62). Multivariable analysis confirmed the significance of all independent factors. In univariable analysis, *post-treatment factors* associated with improved survival were surgery (HR = 0.29; 95%CI: 0.12–0.68) and negative resection margin after surgery (HR 0.41; 95%CI: 0.24–0.71). Metastasis was associated with decreased overall survival (HR 2.3; 95%CI: 1.51–3.59). Multivariable analysis confirmed the independent association of all factors (Table 2).

#### 3.5.2. Metastases

In univariate Fine and Gray competing risk regression models, factors associated with higher risk of metastases were age (HR 0.99; 95%CI: 0.97–1.0) and deep-seated soft tissue tumors (HR 1.2; 95%CI: 0.2–8.0). Head and neck (HR 0.4; 95%CI: 0.16–0.80) and cutaneous tumors (HR 0.19; 95%CI: 0.02–1.6) were associated with a lower risk of metastasis. In multivariate analysis, only cutaneous tumors emerged as a prognosticator for lower risk of metastasis (Table 3).

#### 3.5.3. Local Recurrence

Univariable Fine and Gray competing risk regression model identified secondary AS (HR 1.80; 95%CI: 1.0–3.1), breast AS (HR 2.0; 95%CI: 1.1–3.4), metastasis at diagnosis (HR 0.13; 95%CI: 0.02–0.9), surgery (HR 19; 95%CI: 2.7–137), and marginal resection (HR 4.6; 95%CI: 1.4–15) as risk factors for local recurrence. Only surgery emerged as independent factors associated with lower risk of local recurrence in multivariable analysis (Table 4).

#### 3.5.4. Patients with Localized Disease

With the aim of reducing the risk of confounding, we analyzed the same factors exclusively in patients with localized disease who underwent surgery (n = 106). Cutaneous tumors (HR 0.03; 95%CI: 0.003–0.25) and negative margin (HR 0.45; 95%CI: 0.2–0.8) were associated with improved overall survival. Metastasis (HR 2.2; 95%CI: 1.4–3.5) was associated with decreased overall survival (Appendix D). Despite increasing age, no factors were associated with a risk of local recurrence (Appendix E). Cutaneous and deep seated tumors were associated with higher risk of metastasis (Appendix F).

#### 3.5.5. Patients with Distant Metastasis

Forty-six (26%) patients were not offered surgery. Oncological chemotherapy (HR 0.44; *p* < 0.01) and radiotherapy (HR 0.29; *p* < 0.001) were the only factors associated with improved overall survival despite dismal 1- and 5-year overall survival of 38% (95%CI: 24%–52%) and 2% (95%CI: 0%–7%), respectively (Appendix G).

## 4. Discussion

Our long-term follow-up study demonstrated overall survival after 5 and 10 years to be 28% and 19%, respectively. We found no difference in overall survival between primary and secondary AS. The risk of local recurrence was 16% and 30% after 1 and 5 years, respectively. The risk of metastasis was 29% and 43% after 1 and 5 years, respectively. Age > 60 years and metastasis were independently associated with worse overall survival. Surgery, negative margins and cutaneous tumors were independently associated, with improved overall survival.

Soft tissue sarcomas are, in general, referred as being deep-seated when located below the fascia [10,11,13,14]. After meticulous review off all pathology reports, our pre-liminary results demonstrated significant differences with regard to overall survival, the risk of metastasis, and local recurrence when cutaneous tumors invaded subcutaneous tissue. Even microscopic subcutaneous invasion demonstrated significantly different pathways. We, therefore, re-defined cutaneous tumors as being located above the fascia. Cutaneous and deeper-seated AS are generally distinguished by subfascial in-growth [10,11,13,14]. We believe that our definition should be taken into consideration in the clinical setting and future studies. Despite a significant difference in overall survival of patients with cutaneous and deep-seated tumors, the overall 5-year survival rate remains poor in those with cutaneous tumors (42%; CI95% 21–57). This is likely caused by the predominant location of cutaneous tumors at the head and neck where technical and anatomic issues challenge curative surgery.

Our findings demonstrated poor overall survival compared to other series including AS in all anatomic locations. Survival rates range between 33% and 59% [5,6,10,11,13,14,15,20,21]. The superior overall survival in previous studies is most likely a reflection of short-term follow-up (range: 13 months–1.9 years) in single center studies [10,11,13]. Furthermore, Lahat et al. [5] solely included patients who had complete excision of a tumor in their survival analysis, and Sinnamon et al. (OS 39.7%) [15] only included patients who underwent surgery. Albores-Saavedra et al. [21] solely included cutaneous ASs from the SEER database, which is reflected in a 5-year overall survival of 51%. In a small single center, Abraham et al. [6] included patients with localized disease yielding a 5-year overall survival of 60%. The high overall survival reported by Abraham et al. [6] is likely explained by a very selected cohort of primarily cutaneous tumors. In a nationwide study including patients with metastasis, Weidema et al. [14] demonstrated an overall 5-year survival of 21.9%, which is comparable to our results. However, despite a nationwide cohort Weidema et al. were only able to include 479 cases of 1125 reported cases with AS or hemangioendothelioma [14], yielding a less representative cohort. Furthermore, Weidema and colleagues demonstrated significantly improved overall survival in patients with secondary AS. Differences in the pathway and overall survival rate between primary and secondary is still debated. In a large SEER database cohort, Yin et al. [22] compared primary and secondary breast AS. They found significantly worse overall survival in secondary AS, and secondary AS demonstrated a nearly two-fold increased risk of death in univariate analysis (HR = 1.89), although this was not confirmed when adjusted in multivariate analysis. Nevertheless, the majority of previous studies report significant or tendencies toward improved overall survival in patients with primary AS [6,10,11,22,23,24]. We found a tendency towards improved overall survival in patients with primary AS, although not significant (*p* = 0.2).

In order to overcome the lack of evidence-based management, prognostic factors are continuously being explored. Numerous studies report tumor size as an individual prognostic factor to predict survival. However, how measurement of tumor size is conducted is very poorly described [2,6,11,20,25,26]. Indeed, even with modern digital imaging of AS, findings of tumor size vary as a result of the heterogeneity, hence complicating precise measurement [27]. During review of all pathology reports, we found descriptions of ill-defined tumors, without clear borders, and, at times, with local satellite lesions. These findings are also described in previous reports [2,21,28]. Thus, tumor size is to be considered unreliable as a prognosticator in the clinical setting. We suggest that this factor is disregarded in the clinical setting and in future studies.

Surgery has, over time, remained an independent prognostic factor with regard to overall survival in several studies and reviews [8,10,14,20,23,25]. Surgery emerged as an independent prognosticator for improved overall survival in our cohort. Nevertheless, as surgery, in general, is not offered to patients with disseminated disease, we suggest that our results, to some extent, are biased by selection among patients who were offered surgery. This is supported by our findings that surgery had no advantageous effect of risk of metastasis or local recurrence despite resection with negative margin, even in patients with localized disease. Reports of resection margin are conflicting in the literature and comparison of studies is challenged by a broad variety among the sample size, tumor location, and follow-up [4,5,8,13,15,20,29,30,31]. When analyzing our entire cohort, we found negative margins to be independently associated with improved overall survival. However, eleven patients who were offered surgery had metastasis at the time of diagnosis and the vast majority of patients with a marginal resection margin were located in the head and neck, where the ability to obtain negative margins are well-known to be challenged [9,15,32]. To avoid possible confounding we sought to analyze patients with localized disease who were offered surgery. Negative margins retained significance with regard to overall survival, yet no association with lower risk of recurrence or metastases was demonstrated. The lacking effect of negative margins is confirmed by a grave overall 5-year survival rate of 43% in patients with localized disease and negative margins, which is considerably lower compared to the overall survival in soft tissues sarcomas in general [33]. Sinnamon et al. [15] found marginal margins to be a prognosticator for worse overall survival. However, they only included patients with cutaneous and soft tissue tumors, and excluded patients who died within 90 days of surgery, which could affect the outcome. Fury et al. [13] also demonstrated improved overall survival in patients with negative margins, although margins did not affect local recurrence. However, the results are based on a single Center study with short follow-up. The lacking prognostic effect of negative margins indicates the severely aggressive ability of AS to metastasize, most likely early in the disease pathway.

In concordance with some previous reports, our study demonstrated a higher risk of metastasis compared to local recurrence [11,31,34]. Indeed, the mean time to metastasis was shorter than the mean time to local recurrence, even though only pathologically verified metastasis was included, hence, inevitably underestimating the incidence. This tendency is also demonstrated in previous studies, even with shorter follow-up [5,6,11,31]. Although most metastases in the present cohort were located in the lung, the distribution demonstrated a high incidence of lymph node metastasis and retroperitoneal metastases Similar to previous reports, the distribution in the present cohort represents a broad and unpredictable appearance [8,13,34], indicating a need for and lack of systemic disease control. Current recommendations of oncological treatment are based on early reports demonstrating the advantage of adjuvant radiotherapy [9,35]. However, these studies are single center studies based on small historical sample sizes, and Pawlik et al. [9] only included scalp AS. Despite tendencies towards the effect of local control [6,10,11,36], the recent literature does not confirm the advantage of adjuvant RT with regard to overall survival [10,11,13,20,25]. Viewing the literature, it appears that adjuvant RT is more advantageous in head and neck AS where the negative margin is challenged [8]. Nonetheless, the effect and modality of oncological therapy is, at present, debated and remains inconclusive [12]. This is supported by the lack of effect on overall survival, metastasis, and local recurrence after both adjuvant and palliative oncological treatment in present cohort. Neither chemotherapy nor radiotherapy emerged as independent prognosticators improving overall survival, risk of metastases, or recurrence. Conversely, chemotherapy was associated with a higher risk of metastasis, which is most likely a reflection of the severe disease pathway in patients who were offered chemotherapy. These results resembles treatment outcomes in osteosarcoma before the introduction of Adriamycin.

Despite the completeness of the study, there are a few inherent limitations due to the nature of the design. Nevertheless, since the study comprises a consecutive cohort, the risk of selection bias is low. Although all pathologists in Denmark use standardized diagnostic criteria and analysis, misclassification and information bias are risks in registry studies. As mentioned, the incidence of metastasis is inevitably underestimated, since we exclusively included patients with pathologically verified metastasis. However, due to the mean age of the cohort and, thus, higher risk of other cancers or metastasis from previous cancers, we chose this methodology.

## 5. Conclusions

Angiosarcoma is a highly aggressive soft tissue sarcoma primarily affecting women and predominantly located at extremities and breast. The risk of metastasis is higher than local recurrence, and metastasis occurs earlier than local recurrence. Tumors with subcutaneous invasion predispose a significant higher mortality and should be considered as deep-seated. Adjuvant oncologic treatment did not benefit the overall survival, risk of metastasis, or local recurrence, even in localized disease. Cutaneous tumors, surgery, and negative margin are favorable prognostic factors for overall survival yet without benefit risk of local recurrence and metastasis, resulting in a grave overall survival.

## Figures and Tables

**Figure 1 cancers-16-01834-f001:**
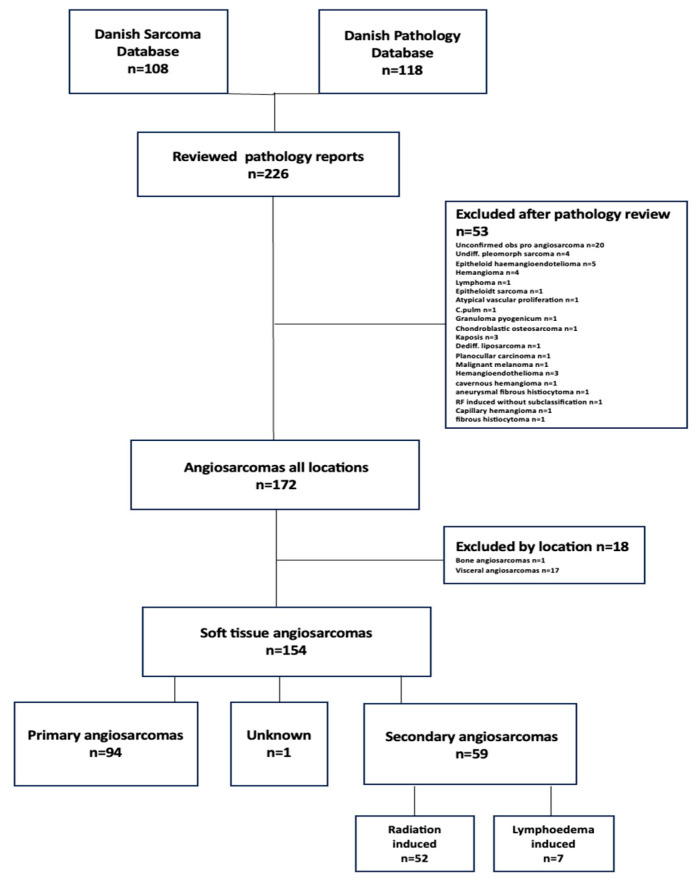
Inclusion of patients. Flowchart illustrating the identification of the cohort and relations between primary and secondary AS.

**Figure 2 cancers-16-01834-f002:**
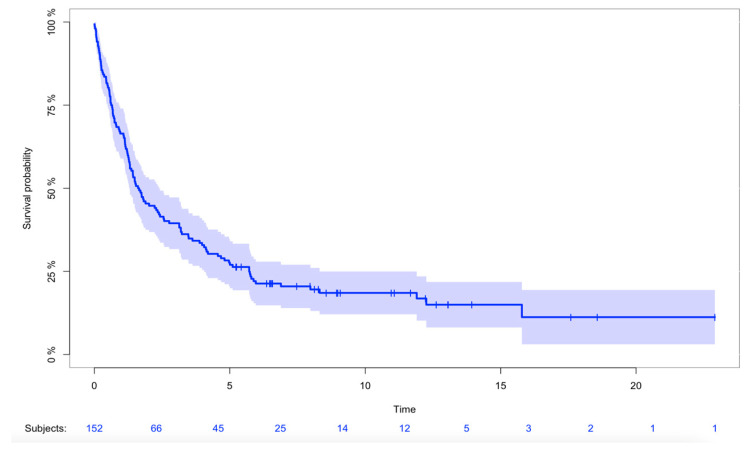
Kaplan–Meier for overall survival of all patients. Kaplan–Meier survival curve with 95%CI demonstrating the probability of overall survival in all patients from 2000 to 2017 (n = 152).

**Figure 3 cancers-16-01834-f003:**
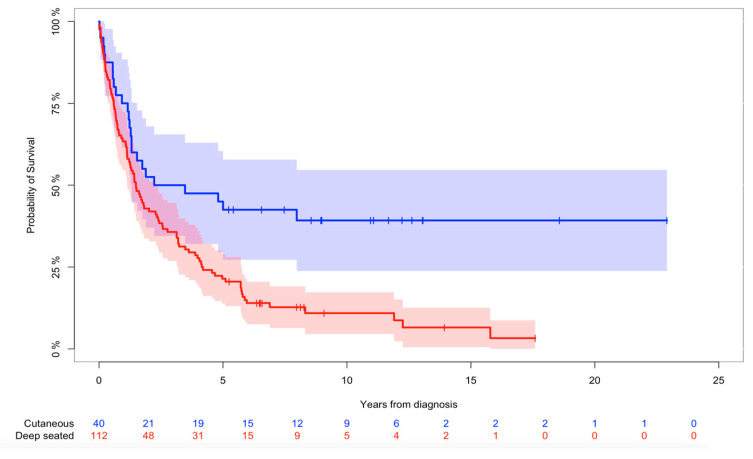
Kaplan–Meier for overall survival of all patients. Kaplan–Meier analysis of overall survival in patients with cutaneous (n = 40) and deep-seated tumors (n = 112) with 95%CI. Log-rank test demonstrated statistically significant difference (*p* = 0.001) in probability of overall survival between cutaneous (blue line) and deep-seated tumors (red line).

**Figure 4 cancers-16-01834-f004:**
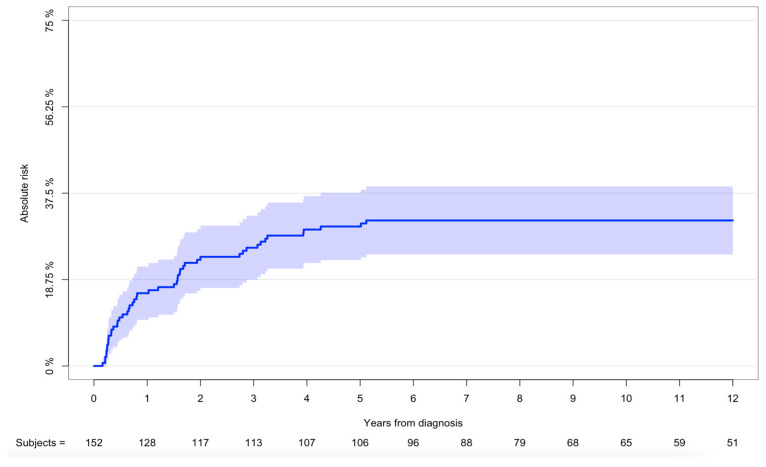
Risk of recurrence of all patients. Cumulative incidence with 95% CI of risk of recurrence—all patients from 2000 to 2017 (n = 152).

**Figure 5 cancers-16-01834-f005:**
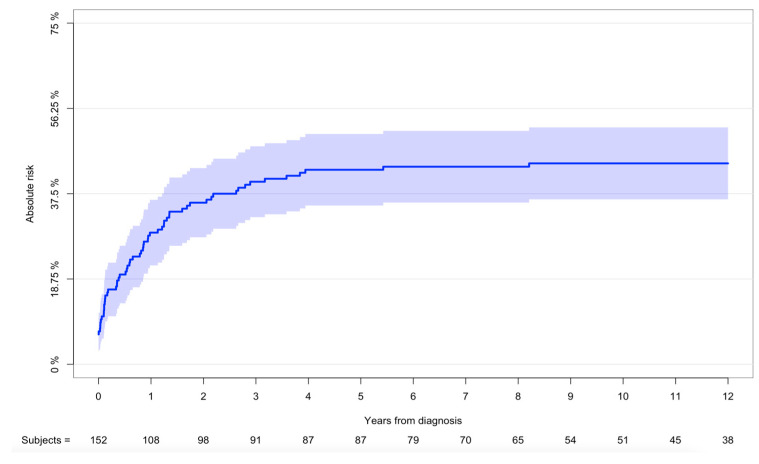
Risk of metastasis of all patients. Cumulative incidence with 95%CI of risk of metastasis—all patients from 2000 to 2017 (n = 152).

**Table 1 cancers-16-01834-t001:** Baseline characteristics of included patients.

	Primary n = 94 (%)	Secondary n = 59 (%)	Missing n = 1 (%)	Total n = 154	*p*-Value
**Gender**					
*Male*	52 (55)	4 (7)		56 (36)	
*Female*	42 (45)	55 (93)	1 (100)	97 (63)	<0.001 *
**Age**					
*Mean (SD)*	65 (17)	68 (14)	74	66 (16)	0.1 #
**Survival**					
*Alive*	15 (16)	12 (20)		27 (18)	
*Not alive*	78 (83)	46 (78)		124 (81)	
*Unknown*	1 (1)	1 (2)		2 (1)	0.6 *
**Metastasis at diagnosis**					
*Yes*	16 (17)	4 (7)		20 (14)	
*No*	78 (83)	55 (93)	1 (100)	133 (86)	0.1 *
**Location**					
*Skin*	23	17		40	0.7
*Deep seated*	67	42	1	110	1
*Unknown*	3	0	-	3	-
**Site**					
*Head and neck*	32	0		32	<0.001
*Trunk*	12	15		28	0.07
*Breast*	10	34	1	44	<0.001
*Extremities*	37	10		47	0.006
*Unknown*	3	0		3	-
**Chemotherapy**					
*Adjuvant*	6	3		9	1
*Palliative*	29	16		45	0.9 *
**Radiotherapy**					
*Adjuvant*	23	5		28	0.03 *
*Palliative*	20	9		29	0.6 *

# student unpaired *t*-test, * Chi^2^ test.

**Table 2 cancers-16-01834-t002:** Univariate and multivariable Cox Proportional Hazard modeling of factors associated with overall survival—all patients (n = 154).

	Univariate Hazard Ratio Pre-Treatment (95%CI)	*p*	Multivariable Hazard Ratio, Preoperative Model (95% CI)	*p*	Multivariable Hazard Ratio, Post-Treatment Model (95% CI)	*p*
**Age, years**						
< 60	Ref.					
>60	1.75 (1.18–2.6)	<0.01	1.94 (1.29–2.89)	<0.01		
**Gender**						
Male	Ref.					
Female	1.28 (0.78–1.85)	0.2				
**Origin**						
Secondary AS	Ref.					
Primary AS	1.18 (0.82–1.71)	0.4				
**Depth**						
Cutaneous	0.48 (0.31–0.76)	<0.001	0.18 (0.05–0.62)	<0.01		
Deep seated	1.82 (1.2–2.8)	<0.001				
**Site**						
Head/neck	0.83 (0.53–1.3)	0.4				
Trunk	1.43 (1.0–2.2)	0.1				
Breast	0.77 (0.5–1.1)	0.2				
Extremities	1.1 (0.76–1.62)	0.6				
**Metastasis at diagnosis**	3.41 (2.1–5.6)	<001	3.13 (1.87–5.24)	<0.001		
None	Ref.					
**Surgery**	0.22 (0.15–0.34)	<001			0.29 (0.12–0.68)	<0.01
None	Ref.					
**Margin**						
R0/negative	0.30 (0.17–0.45)	<0.001			0.41 (0.24–0.71)	<0.01
R1/positive	Ref.					
**Recurrence**	1.34 (0.51–1.09)	0.1				
None	Ref.					
**Oncologic ** **treatment**						
Radiotherapy *	1.17 (0.81–1.69)	0.4				
None	Ref.					
Chemotherapy **	1.27 (0.87–1.84)	0.2				
None	Ref.					
**Metastasis**	1.74 (1.2–2.5)	<0.01		<0.001	2.3 (1.51–3.59)	<0.001
None	Ref.					

* adjuvant and palliative radiotherapy, ** adjuvant and palliative chemotherapy.

**Table 3 cancers-16-01834-t003:** Univariable and multivariable Fine and Gray competing risk regression models of factors associated with metastasis—all patients (n = 154).

	Univariate Hazard Ratio Pre-Treatment (95%CI)	*p*	Multivariable Hazard Ratio, Preoperative Model (95% CI)	*p*	Multivariable Hazard Ratio, Pos-Treatment Model (95% CI)	*p*
**Age, years**						
	0.99 (0.97–1.0)	0.02				
**Gender**						
Male	Ref.					
Female	1.01 (0.6–1.7)	0.9				
**Origin**						
Secondary AS	0.8 (0.5–1.4)	0.4				
Primary AS	Ref.					
**Depth**						
Cutaneous	0.13 (0.05–0.38)	<0.001	0.19 (0.02–1.6)	0.13		
Deep seated	5.3 (2.2–13.0)	<0.001	1.2 (0.2–8.0)	0.8		
**Site**						
Head/neck	0.4 (0.16–0.80)	0.01	0.6 (0.3–1.4)	0.3		
Trunk	0.80 (0.4–1.5)	0.5				
Breast	1.6 (1.0–2.5)	0.1				
Extremities	1.3 (0.8–2.2)	0.3				
**Surgery**						
Surgery	1.5 (0.8–3.1)	0.2				
None	Ref.					
**Margin**						
R0/negative	0.9 (0.5–1.8)	0.8				
R1/positive	Ref.					
**Recurrence**	1.1 (0.7–1.8)	0.6				
None	Ref.					
**Oncologic ** **treatment**						
Radiotherapy *	1.6 (1.0–2.6)	0.05			1.3 (0.8–2.1)	0.3
None	Ref.					
Chemotherapy **	2.6 (1.6–4.2)	<0.001			2.4 (1.4–4.0)	<0.001
None	Ref.					

* adjuvant and palliative radiotherapy, ** adjuvant and palliative chemotherapy.

**Table 4 cancers-16-01834-t004:** Univariable and multivariable Fine and Gray competing risk regression models of factors associated with recurrence—all patients (n = 154).

	Univariate Hazard Ratio Pre-Treatment (95%CI)	*p*	Multivariable Hazard Ratio, Preoperative Model (95% CI)	*p*	Multivariable Hazard Ratio, Pos-Treatment Model (95% CI)	*p*
**Age, years**						
	1.02 (1.0–1.03)	0.06	1.02 (1.0–1.03)	0.04		
**Gender**						
Male	Ref.					
Female	0.9 (0.5–1.7)	0.8				
**Origin**						
Secondary AS	1.8 (1.0–3.1)	0.05	1.1 (0.6–2.1)	0.7		
Primary AS	Ref.					
**Depth**						
Cutaneous	0.9 (0.5–1.7)	0.7				
Deep seated	1.3 (0.7–2.4)	0.5				
**Site**						
Head/neck	0.7 (0.3–1.5)	0.4				
Trunk	0.4 (0.1–1.1)	0.06				
Breast	2.0 (1.1–3.4)	0.02	1.8 (1.0–3.4)	0.07		
Extremities	1.2 (0.7–2.2)	0.6				
**Metastasis at diagnosis**	0.13 (0.02–0.9)	0.04	0.1 (0.02–1.1)	0.06		
None	Ref.					
**Surgery**	19 (2.7–137)	<0.01			10.5 (2.7–40)	<0.001
None	Ref.					
**Margin**						
R0/negative	4.6 (1.4–15)	0.01			2.9 (0.9–10.0)	0.09
R1/positive	Ref.					
**Oncologic** **treatment**						
Radiotherapy *	1.1 (0.6–2.0)	0.8				
None	Ref.					
Chemotherapy **	1.6 (0.9–2.9)	0.1				
None	Ref.					
**Metastasis**	1.4 (0.8–2.4)	0.3				
None	Ref.					

* adjuvant and palliative radiotherapy, ** adjuvant and palliative chemotherapy.

## Data Availability

The data presented in this study have restrictions due to patient confidentiality and are, therefore, not publicly available. However, they can be made available on reasonable request from the corresponding author.

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
