# Peer review of "A Population-Based Long-Term Follow-Up of Soft Tissue Angiosarcomas: Characteristics, Treatment Outcomes, and Prognostic Factors"

_cancers, 2024, doi:10.3390/cancers16101834_

Round 1

Reviewer 1 Report

Comments and Suggestions for Authors

The topic is interesting, but the series is extremely heterogeneous. I would concentrate on STS. Cutaneous and visceral angiosarcomas should be excluded as different entities.

Also, the study is disorganized as inclusion/exclusion criteria are not clear.

Moreover, the paper is disorganized and difficult to follow.

Moreover, it seems that also bone sarcomas were included. Thus, this is extremely confusing.

Please detail in the title about STS angiosarcoma.

Was any case excluded because of incomplete information?

A minimum follow up of 1 years is mandatory (0 days FU cases must be excluded)

Also, please email about radio/chemotherapies regimens.

How were surgical margins assessed?

As the study period is very long, I would suggest to verify whether prognosis ameliorated over years, as chemotherapy regimens modified.

Kaplan Meier plots: patients should be censored at last follow up available

Conclusions are not supported completely. 

References: many papers missing.

Comments on the Quality of English Language

Many grammar and syntax errors

Author Response

1. The topic is interesting, but the series is extremely heterogeneous. I would concentrate on STS. Cutaneous and visceral angiosarcomas should be excluded as different entities.

Answer:

Two primary angiosarcomas of bone has now been excluded, leaving only soft tissue tumors.

2. Please detail in the title about STS angiosarcoma.

Answer:

Besides from two excluded bone angiosarcoma cases, the entire population based cohort comprises soft tissue sarcomas and we therefor think the present title is appropriate.

3. Was any case excluded because of incomplete information?                                                              

Answer

All information of interest were available in our national registers. No cases were excluded due to missing information. Missing values are reported.

4. A minimum follow up of 1 years is mandatory (0 days FU cases must be excluded)                               

The aim of our study is to make the first population based long-term evaluation in Denmark in patients with angiosarcoma. The aim of the study is to evaluate the morbidity and mortality of the disease. The whole cohort is the population of interest. Patients with follow up less than 12 months is not a result of loss to follow up. Patients with follow up of less than 12 months are patients who died within follow-up as a result of a rapidly developing disease course. We find the short term survival in patients with disseminated disease of highly interest as a reflection of the current lack of disease control and the aggressive behavior of the disease. However, for the purpose of this interest we have supplemented our results with overall survival in patients with more than 12 months (appendix B).

5. Also, please email about radio/chemotherapies regimens.                                                               

Answer

All patients were treated with either Taxol or Doxorubicin.

5. How were surgical margins assessed?                                                                                           

Answer

All surgical margins were assessed after meticulous review of all pathology reports. In doubt pathology was assessed by a board-certified sarcoma pathologist (co-author)

6. As the study period is very long, I would suggest to verify whether prognosis ameliorated over years, as chemotherapy regimens modified.                                                         

Answer

Oncological data was obtained from patient files. After viewing all patient files we can conclude that all patients were treated with either Taxol or Doxorubicin

7. Kaplan Meier plots: patients should be censored at last follow up available                                      

Answer

We aimed to define the probability of overall survival. Exact date of death of all patients are registered and patients are therefor censored accordingly.  

8. References: many papers missing.                                                                                              

Anser

We have added relevant reference

Reviewer 2 Report

Comments and Suggestions for Authors

With this study, authors characterized a cohort of 143 patients with angiosarcoma, a rare type of soft tissue sarcoma; Population of patients is well described and all data available were systematically analyzed by authors. 

Except the Figure 1 that could be better presented in a more readable manner, this study is clear, comparison with other communications are made and no obvious bias is observed. 

Author Response

Except the Figure 1 that could be better presented in a more readable manner, this study is clear, comparison with other communications are made and no obvious bias is observed. 

Answer

The layout of Figure 1 has now been changed to a hopefully more readable manner. We will be happy to edit further if necessary.

Reviewer 3 Report

Comments and Suggestions for Authors

Angiosarcoma are very rare malignancies; available epidemiological data are therefore limited. This analysis of Danish registry data is relevant as it contributes to observations collected by other authors in the past. Basically, the article confirms the prognostic factors and the poor outlook of this malignancy reported in the literature. Survival data are analysed according to the location (cutaneous vs. deep-seated). Other important factors are also reported such as surgical outcomes, metastasis or recurrence. The figure and tables are clear.

Comments to the authors:

The better overall survival of cutaneous tumours may also be explained by the fact that they are better amenable to surgery than deep-seated, infiltrating tumours. Surgery and location are thus not completely independent prognostic factors.

The authors are encouraged to comment whether there is an increase of the incidence of angiosarcoma over the years as has been reported very recently for angiosarcoma in the US (https://jamanetwork.com/journals/jamanetworkopen/fullarticle/2817478) and for other malignancies.

Author Response

The authors are encouraged to comment whether there is an increase of the incidence of angiosarcoma over the years as has been reported very recently for angiosarcoma in the US (https://jamanetwork.com/journals/jamanetworkopen/fullarticle/2817478) and for other malignancies.

Answer

We found a tendency although not significant. These data has now been added as supplementary material (appendix A)

Reviewer 4 Report

Comments and Suggestions for Authors

In a Danish follow-up study, 173 consecutive patients diagnosed with rare angiosarcoma were analyzed to assess treatment outcomes and prognostic factors. The study affirmed the presence of low long-term survival rates, with adjuvant treatment failing to demonstrate any beneficial effects. The authors identified several favorable factors, including surgery with negative margins and cutaneous tumors.

While the authors endeavored to adhere to the latest WHO guidelines for tumor classification, they proposed a future standard to classify tumors infiltrating subcutaneous tissue as deep-seated soft tissue tumors. The reviewer commends this informative study on a rare tumor type; the analysis and conclusions drawn are well-supported by the results.

Author Response

On behalf of all authors, we thank this reviewer for the appreciation of the study.

Round 2

Reviewer 1 Report

Comments and Suggestions for Authors

Despite the Authors' efforts, they were not able to address appropriately to most of my previous concerns. Thus, the paper is still not suitable for the publication.

Author Response

All authors appreciate for taken your time to revise this paper.

With respect for the aim of the study, we have made the best effort possible to accommodate your suggestions.